# Global economic costs of alien birds

**Thomas Evans** [ID]<sup>1</sup>*, **Elena Angulo** [ID]<sup>1,2</sup>, **Corey J. A. Bradshaw**<sup>3,4</sup>, **Anna Turbelin**<sup>1</sup>, **Franck Courchamp**<sup>1</sup>

**1** Ecologie Systématique et Evolution, Université Paris-Saclay, Gif-sur-Yvette, France, **2** Estación Biológica de Doñana (CSIC), Seville, Spain, **3** Global Ecology | *Partuyarta Ngadluku Wardli Kuu*, College of Science and Engineering, Flinders University, Adelaide, South Australia, Australia, **4** ARC Centre of Excellence for Australian Biodiversity and Heritage, EpicAustralia.org.au, Australia

* thomgevans@gmail.com

**Data Availability Statement:** All relevant data are within the paper and its Supporting Information files, or freely available from DOI: https://doi.org/10.6084/m9.figshare.12668570.v5.

**Funding:** Our work was supported by the French National Research Agency [ANR-18-EBI4-0004-07,

## Abstract

The adverse impacts of alien birds are widespread and diverse, and associated with costs due to the damage caused and actions required to manage them. We synthesised global cost data to identify variation across regions, types of impact, and alien bird species. Costs amount to US$3.6 billion, but this is likely a vast underestimate. Costs are low compared to other taxonomic groups assessed using the same methods; despite underreporting, alien birds are likely to be less damaging and easier to manage than many other alien taxa. Research to understand why this is the case could inform measures to reduce costs associated with biological invasions. Costs are biassed towards high-income regions and damaging environmental impacts, particularly on islands. Most costs on islands result from actions to protect biodiversity and tend to be low and one-off (temporary). Most costs at mainland locations result from damage by a few, widespread species. Some of these costs are high and ongoing (permanent). Actions to restrict alien bird invasions at mainland locations might prevent high, ongoing costs. Reports increased sharply after 2010, but many are for local actions to manage expanding alien bird populations. However, the successful eradication of these increasingly widespread species will require a coordinated, international response.

## Introduction

Species that have been introduced through human endeavour to regions that they would not normally inhabit are known *inter alia* as 'alien' species. Some of those species have damaging impacts on wildlife and people–they are further defined as 'invasive alien' species [1]. There are more than 400 alien bird species with self-sustaining populations worldwide [2], and some are considered invasive [3]. In recent years, research has improved our understanding of their impacts. Some studies have focussed on specific alien bird species (e.g., predation of native seabirds by the barn owl *Tyto alba* in Hawai'i [4]; hybridisation between the Chinese hwamei *Garrulax canorus* and the native Taiwan hwamei *Garrulax taewanus* in Taiwan [5]; and competition for nest-cavities between the rose-ringed parakeet *Alexandrinus krameri* and the native Eurasian nuthatch *Sitta europaea* in Belgium [6]). Other studies have reviewed or assessed the impacts of all alien bird species from an entire order of birds, such as alien

2018]; the BNP-Paribas Foundation Climate Initiative [2014-00000004292, 2014]; the AXA Research Fund Chair of Invasion Biology [2019]; and the AlienScenario project funded by BiodivERsA and Belmont-Forum call 2018 on biodiversity scenarios [I 4011-B32, 2018]. Our funders had no role in study design, data collection and analysis, decision to publish, or preparation of the manuscript.

**Competing interests:** The authors have declared that no competing interests exist.

Psittaciformes (parrots) in the USA and UK [7], Europe [8], and worldwide [9]. Another study has identified the negative impacts of alien birds caused by a specific type of impact (predation) on small islands [10]. Other studies have focussed on alien birds as an entire taxonomic class, including global reviews of their impacts [11–13], and assessments that have quantified and categorised their environmental and socio-economic impacts by their severity and type [14, 15], and that have identified factors that make native species vulnerable to these impacts [16].

This body of research shows that the impacts of alien birds are widespread and diverse, affecting biodiversity and people across the globe in many different ways. These impacts are often associated with economic costs, either directly due to the damage they cause, or indirectly due to expenditures associated with their management. For example, the Egyptian goose *Alopochen aegyptiaca* causes damage by grazing crops in the Netherlands [17], while management costs have been incurred when eradicating the common myna *Acridotheres tristis* from the Seychelles to mitigate its negative impacts on native wildlife [18].

Although cost estimates have been produced for some alien bird species, a global review of their reported economic costs has yet to be completed. Yet such a review could improve understanding of how and where alien birds generate economic costs, and how these costs vary in scale across regions, types of impacts, and species. This information could inform management interventions to reduce the financial burden placed on human societies by alien birds.

*InvaCost* is a living, publicly available database listing the reported costs of alien species worldwide [19]. It incorporates a methodology to standardise historical cost data in different currencies to current values in a single currency (US$ at the 2017 exchange rate). This enables meaningful comparisons of past and current costs incurred among different regions, and associated with different types of alien species and different types of impact. *InvaCost* has been used to assess the economic costs associated with several groups of alien species, including ants [20], bivalves [21], crustaceans [22], herpetofauna [23], terrestrial invertebrates [24], fish [25], and mammals [26]. These studies demonstrate that costs associated with alien species can be enormous (e.g., since 1930, the costs associated with the impacts of 12 alien ant species worldwide are > US$10 billion [20]; since the 1960s, costs associated with the impacts of alien mammals worldwide are > US$450 billion [26]) and that they also tend to be underreported [27, 28]. However, data contained within the *InvaCost* database have yet to be used to assess the costs of alien birds.

In this study, we synthesise data on the reported costs of alien birds across different regions of the world. Based on the results of previous studies, we pose several hypotheses. Because the costs of alien birds have been linked to both their environmental and socio-economic impacts, and these impacts have been reported from many regions of the world [13, 15], we hypothesise that data on costs will be widespread (hypothesis 1). Nevertheless, the costs of alien species tend to be underreported, particularly in low-income regions [29, 30], so costs will be unavailable for many regions (hypothesis 2). The environmental impacts of alien birds are sometimes managed (e.g., recent eradication of house sparrows *Passer domesticus* on Robinson Crusoe Island, Chile [31]), and some costs will therefore be associated with biodiversity conservation. However, alien-species research tends to focus on damaging environmental impacts [30], so more cost data will be available for damaging alien bird species because they are more likely to be managed (hypothesis 3). Nevertheless, the environmental impacts of some alien bird species are difficult to value economically, such as predation of a native bird species by an alien raptor (bird of prey), and some of these impacts have yet to be managed [10]. Therefore, data on costs will be unavailable for some regions where severe environmental impacts are reported (hypothesis 4). The reported management costs associated with the only other group of terrestrial vertebrate species assessed using *InvaCost* (mammals) tend to be much lower than their

damage costs [26], and the same pattern is likely to be true for birds, particularly because many of their impacts are not managed (hypothesis 5). Many alien bird species are likely to have minor environmental [32, 33] and socio-economic impacts [15], and therefore their economic costs (which are associated with these impacts) will be low compared to many other taxonomic groups of alien species assessed using *InvaCost* (hypothesis 6).

## Materials and methods

### Data

We extracted data on the economic costs of alien birds from the *InvaCost* database [34], which contains > 13,000 records of economic costs associated with alien species that have been gathered through literature searches (see Diagne *et al.* 2020 [19] for a full description of the *InvaCost* database and associated methods). We reviewed the 389 entries relating to alien birds, removing 44 records considered unreliable due to a lack of information or discrepancies, and adding six new cost records identified since the publication of the latest version of the database. Thus, our review is based on 351 cost records. We used the *expandYearlyCosts* function of the `invacost` R package [35] to distribute cost records annually over their reported timeframe (e.g., a $60,000 cost incurred between 1999 and 2001 would be transformed to $20,000 for each of the three years during which costs were incurred).

*InvaCost* incorporates a set of descriptors that enable detailed analysis of costs associated with specific attributes of alien species. Using these descriptors, we collected data on: (*i*) the number of cost records/year, (*ii*) the reliability of cost records ('low' or 'high'); (*iii*) the permanence of costs (one-off [temporary] or ongoing [permanent]); and (*iv*) whether costs were observed (realised) or potential (predicted). We then calculated costs for the following categories by summing costs associated with all records: (*v*) alien bird order, (*vi*) alien bird species, (*vii*) country of reported costs, (*viii*) type of incurred costs–'management' or 'damage' (two damage categories: agricultural damage or damage to other assets, including facilities/infrastructure/buildings) or 'mixed' if the data did not distinguish between management or damage costs, and (*ix*) year during which an impact caused costs (summing one-off costs for each year with ongoing costs from previous years).

### Analysis

Most records in the *InvaCost* database described costs associated with a single alien bird species. However, a few cost records (< 3%) were associated with two or three species. In these cases when calculating costs for an alien bird species, we assigned each species the total cost for the record. This is because there was not enough information to assign proportionate costs accurately to each species. For example, these records included costs associated with the management of mixed flocks of Egyptian geese and Canada geese *Branta canadensis*. In other cases, several alien bird species were associated with a cost record, and they were not clearly named, being described as 'exotic birds' or 'introduced birds', for example. These costs were not assigned to specific species, but to a category titled 'diverse/unspecified'.

Over half of all costs were caused by street pigeons *Columba livia forma urbana*. The impacts of this species are diverse and widespread, and associated costs are difficult to calculate accurately. We therefore assigned them a 'low' reliability and categorised them as 'mixed' costs (i.e., both damage and management costs), because accurate information on their nature was not available. Given their low reliability, we analysed cost data with and without costs for street pigeons.

## Results

The total reported costs associated with the impacts of alien birds amounted to approximately US$3.6 billion (US$1.6 billion with costs for street pigeons excluded). Approximately 97% were observed costs, and 3% were potential costs (with street pigeon costs removed, 94% were observed and 6% potential). With costs for the street pigeon excluded (which had a 'low' reliability), most cost records were categorised as 'high' reliability (95%; $n$ = 333).

### Spatial distribution

Costs tended to be reported in high-income regions of the world, such as Australasia, Western Europe, and USA (Fig 1) (see S1 Fig for costs with street pigeons excluded). No cost data were available for many lower-income regions occupied by alien birds (e.g., India, and most of Africa, Southeast Asia, and South America). However, many reports were for costs incurred on islands, including those within low-income regions such as Socotra (Yemen).

Total costs incurred on islands were much lower than those incurred at mainland locations (even when excluding high costs for street pigeons at mainland locations) (Fig 2). Damage to facilities/infrastructure/buildings was reported at mainland locations, but not on islands; damage to agriculture was reported at both mainland and island locations (Fig 2). Management costs were also reported at both mainland and island locations. However, a higher proportion of total costs on islands (i.e., damage and management costs) were associated with management (53%) when compared with mainland locations (6.3% with costs for street pigeons excluded) (Fig 2). All management costs on islands were for schemes to control or eradicate alien birds, with the costs incurred by stakeholders and government agencies responsible for the protection of the environment.

Almost all potential costs were at mainland locations (Australia, Western Europe, and USA), with only one island potentially incurring costs (New Caledonia). These costs included

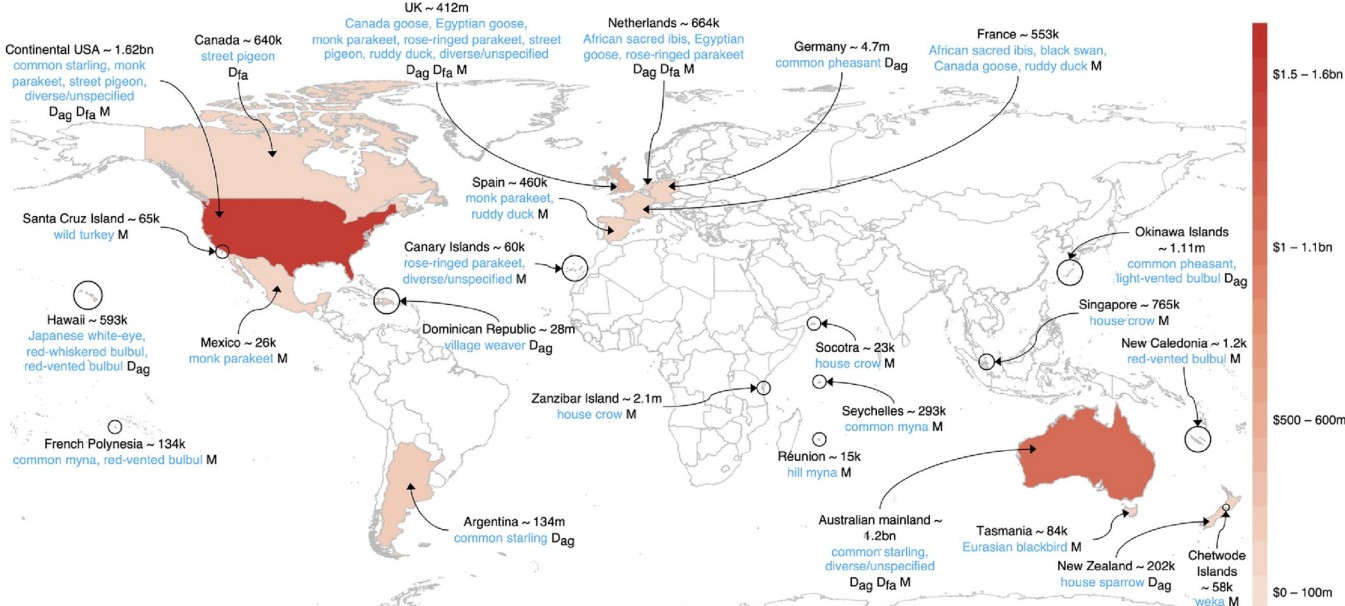

**Fig 1. The spatial distribution of the observed economic costs associated with alien birds (US$, 2017 exchange rate).** k = thousand; m = million; bn = billion. $D_{ag}$ = costs associated with damage to agriculture; $D_{fa}$ = costs associated with damage to facilities/infrastructure/buildings; M = cost associated with management. This map was made with Natural Earth. Free vector and raster map data @ naturalearthdata.com.

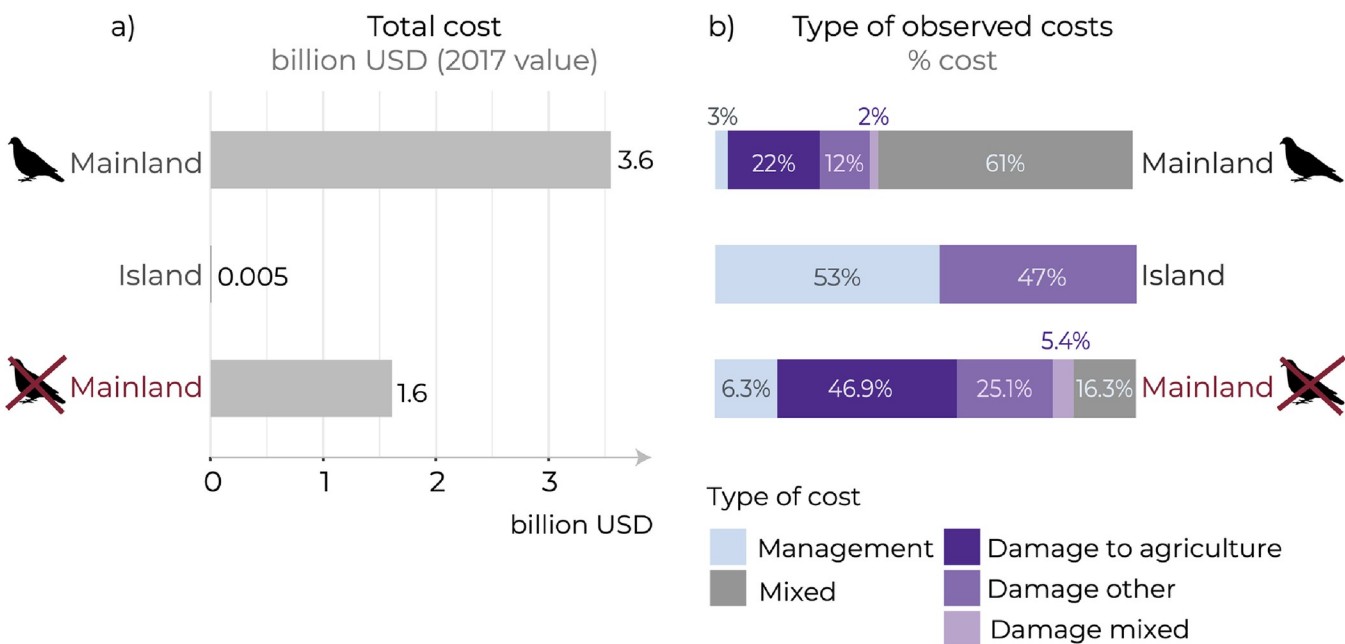

**Fig 2.** (a) total observed costs associated with alien birds incurred at mainland or island locations; (b) proportion of different types of observed costs incurred at mainland or island locations. Costs at mainland locations are shown with and without costs for street pigeons *Columba livia forma urbana*. 'Mixed' = combined 'damage' and 'management' costs (where information was insufficient to separate them). All costs for street pigeons were categorised as 'mixed'. 'Damage mixed' = combined 'damage to agriculture', and 'damage other' costs (where information was insufficient to separate them).

agricultural damage by Canada geese and Egyptian geese in Europe, common starlings *Sturnus vulgaris* in Australia, and red-vented bulbuls *Pycnonotus cafer* in New Caledonia.

## Taxonomic distribution

Reported costs were associated with 22 alien bird species from seven orders (approximately 5% of established alien bird species worldwide) (Fig 3). Columbiformes (pigeons and doves) were associated with approximately 55% of all costs; all were caused by street pigeons. Passeriformes (perching birds) were also associated with high costs, with most caused by common starlings (approximately 4% of all costs; 9% when street pigeon costs are excluded). Over one-third (38%) of observed costs were allocated to the 'diverse/unspecified' category; with costs for street pigeons removed this figure increased to 86%.

## Temporal distribution

Although some cost records were from the last century, most (86%) were from 2000 to the present, with a sharp increase in records after 2010 (63% of reports were from 2010–2019) (Fig 4). However, total cumulative costs year$^{-1}$ have remained approximately stable over this period (approximately US$2.5 billion with costs for street pigeons; US$500 million without), and average cumulative costs year$^{-1}$ tend to be declining (Fig 4).

Excluding costs for street pigeons (all of which are ongoing), approximately 64% of observed costs were one-off (temporary) costs. The remainder (36%) were ongoing (permanent) costs associated with damage and the management of alien birds where their complete removal is unlikely (Fig 5). With costs for the street pigeon included, ongoing (permanent) costs amounted to 70% of all costs (the remaining 30% being one-off (temporary) costs).

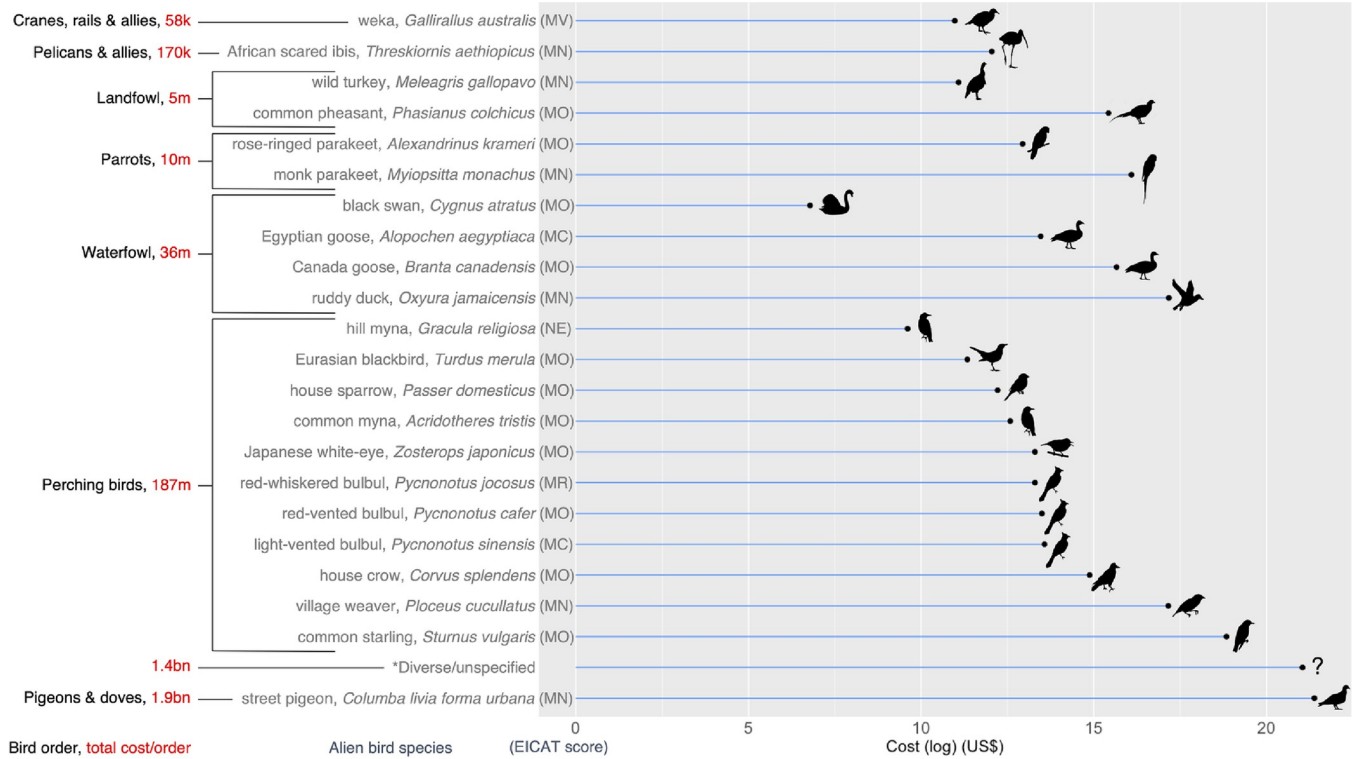

**Fig 3. The observed economic costs (log$_e$) associated with 7 bird orders, 22 alien bird species and *diverse/unspecified species.** Total observed costs for each bird order are provided in red (US$, 2017 exchange rate); k = thousand; m = million; bn = billion. The most severe reported biodiversity impact caused by each species, as categorised using the *Environmental Impact Classification for Alien Taxa* (EICAT) framework, is also provided. EICAT impact categories: MC = Minimal Concern, MN = Minor, MO = Moderate, MR = Major, MV = Massive, NE = Not Evaluated. MO, MR, and MV impacts are considered 'harmful' under EICAT.

## Discussion

The distribution of data on the costs of alien birds is widespread and broadly congruent with the distribution of data describing their environmental and socio-economic impacts [15, 29] (hypothesis 1). More cost data are available in high-income regions of the world such as Australia, Western Europe, and USA (hypothesis 2), most likely because more research on the impacts of alien species tends to occur in these regions [29, 30, 36]. Cost data are unavailable for many low-income regions of the world, as is the case for other groups of alien species such as alien crayfish in Africa [22], alien fish in South America and Africa [25], and alien ants in Eastern Europe, Africa and Southeast Asia [20]. Yet, it is these regions where the impacts of alien bird species might be most problematic because they are more likely to reduce food security [37].

Most alien bird species do not have data describing their costs, and for some this might therefore be because they only occur as aliens in low-income countries, where their costs have not been reported (hypothesis 2). However, another possible explanation is that the environmental and socio-economic impacts of alien birds are often relatively minor [14, 15, 33], and invasion biology research tends to focus on the most damaging alien species [30]. For example, dunnocks *Prunella modularis* were introduced to New Zealand over 150 years ago, and this species is now common and widespread in southern regions [38]. However, it has been categorised as *Data Deficient* (DD) under the *Environmental Impact Classification for Alien Taxa* (EICAT) framework, because there are no data on its impacts from which to complete an

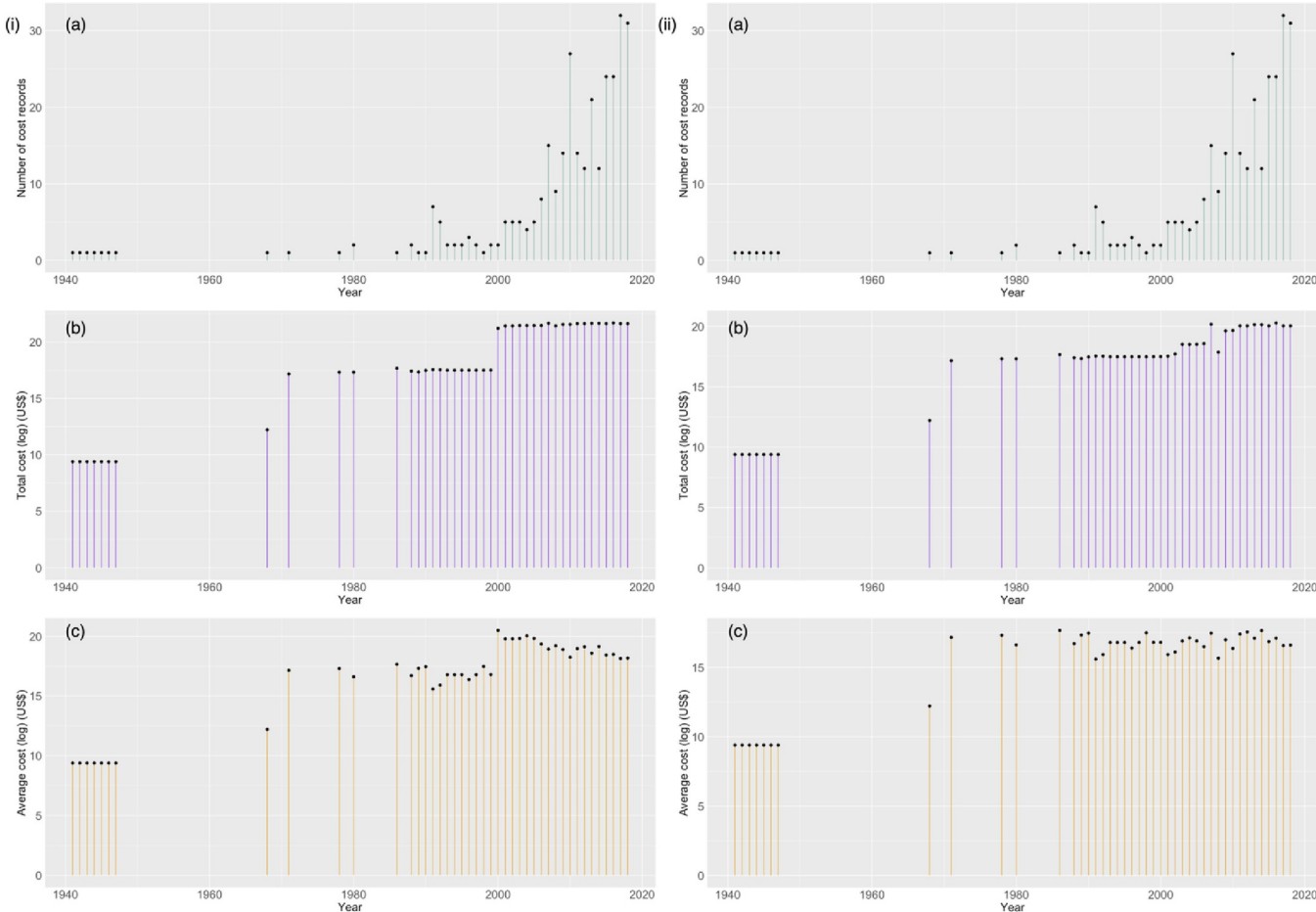

**Fig 4.** (a) number of observed cost records year$^{-1}$ associated with alien birds; (b) total cumulative observed economic costs year$^{-1}$ (log$_e$) associated with alien birds (calculated by summing all one-off (temporary) costs for a year along with all ongoing (permanent) costs for that year and ongoing (permanent) costs for all previous years); (c) average cumulative observed economic costs year$^{-1}$ (log$_e$) associated with alien birds. Costs are shown (*i*) with and (*ii*) without costs for street pigeons.

EICAT assessment [14]. This suggests that the impacts of the dunnock in New Zealand are likely negligible. Therefore, alien bird species with no cost data could be those with minor impacts that do not warrant priority research to estimate their costs, which are probably low (hypothesis 6). This association between minor impacts and data deficiency for alien bird species has been inferred for their environmental impacts [32], socio-economic impacts [15], and monetary costs (using *InvaCost*) [39]. Thus, it is possible that a small proportion of established alien bird species have high economic costs. However, it is also possible that high costs are not reported, even in high-income regions. For example, in Europe, data on the costs associated with alien species are not always accessible [40]. Therefore, some alien bird species with no cost data could in fact have high costs.

Furthermore, the total economic costs of alien birds (US$3.6 billion), over half of which are caused by street pigeons alone, are low relative to costs associated with most other taxonomic groups assessed using *InvaCost* (hypothesis 6). They include terrestrial invertebrates (US$712 billion) [24] (including ants, which alone have costs of US$10.95 billion) [20], mammals (US$462 billion) [26], bivalves (US$63.7 billion) [21], fish (US$37.08 billion) [25], and herpetofauna (US$16.98 billion) [23]. This could be because certain ecological characteristics of birds

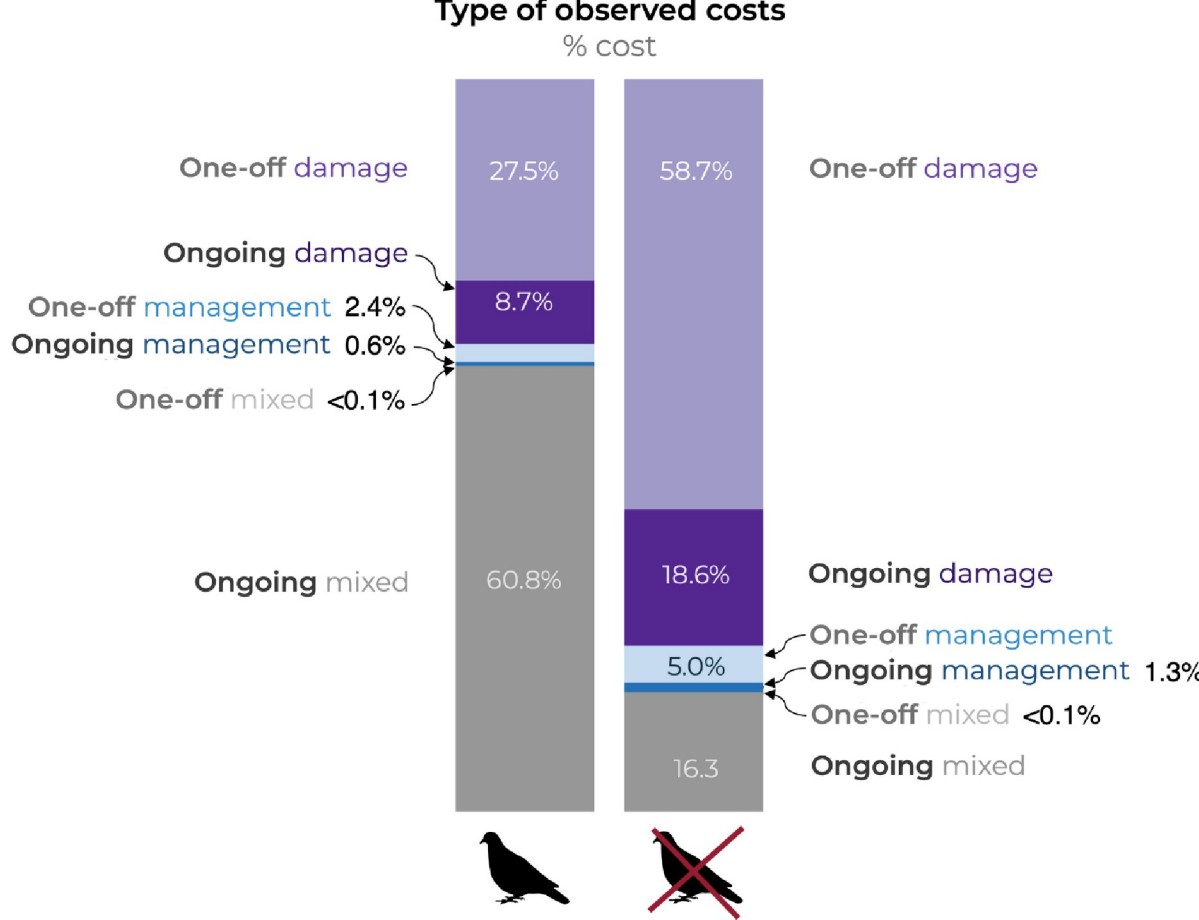

**Fig 5. The proportion of observed costs associated with alien birds that are one-off (temporary) or ongoing (permanent) (with and without costs for street pigeons *Columba livia forma urbana*).** 'Mixed' = combined 'damage' and 'management' costs (where information was insufficient to separate them). All costs for street pigeons were categorised as 'mixed'.

make them easier to manage and their impacts less damaging. For example, birds are more conspicuous compared to some other taxonomic groups such as terrestrial invertebrates, and might therefore be easier to identify for management–multiple alien populations of African black sugar ants *Lepisiota incisa* are thought to have gone undetected in Australia for up to five years before their discovery in 2020, and extensive survey work has been necessary to establish their extent [41]. Furthermore, birds tend to occupy habitats above the ground or water level, unlike ants and crayfish, and they do not physically attach themselves to structures, unlike bivalves. Indeed, damage to drinking water, plant water intake infrastructure, and irrigation systems in North America due to fouling by alien bivalve species has cost at least $10 billion since 1980 [21]. Compared to other groups of alien species, birds might therefore be easier to contain, physically remove, and eradicate, despite being highly mobile. Birds also tend not to alter the structure of the environment profoundly, and therefore their impacts do not often require costly remediation, unlike ants [20] and bivalves [21]. Research to improve our understanding of why alien birds tend to have lower costs than many other alien taxa could inform measures to prevent high costs associated with future biological invasions.

Nevertheless, alien bird species can have severe environmental impacts that are difficult to monetise. Indeed, 58% of all alien bird species assessed as 'harmful' to biodiversity under the

EICAT framework have no reported costs (hypothesis 4). They cause negative impacts through a range of mechanisms including competition, predation, hybridisation, brood parasitism, overgrazing of vegetation, and disease transmission (see S1 Table for details). An example of an alien bird species with severe biodiversity impacts and no cost data is the great horned owl *Bubo virginianus*. Through predation, this raptor has caused the decline of several native bird species and the extirpation of the Marquesas kingfisher *Todiramphus godeffroyi* [42] on Hiva Oa (French Polynesia). There are likely to be many intangible, non-monetisable environmental impacts associated with alien birds that adversely affect nature's contribution to people [43].

Cost data are also congruent with environmental impact data because these costs and impacts are often linked. In these cases, costs are incurred when managing the environmental impacts of alien birds (mainly through their control or eradication). Indeed, approximately 60% of the alien bird species with costs also have environmental impacts defined as being 'harmful' under the EICAT framework [44] (Fig 3). The severity of these impacts is the trigger for their management, and hence costs (hypothesis 3). Because the environmental impacts of alien bird species tend to be more severe on islands [10, 45], many of these management costs are incurred on islands.

Nevertheless, management costs on islands are low relative to management costs at mainland locations. Even excluding the high costs associated with the street pigeon at mainland locations, average management costs/records at mainland locations are four times higher than those on islands (US$363,000 *versus* US$90,000, respectively). This might be because small populations of alien bird species have been eradicated on islands for a low cost (e.g., weka *Gallirallus australis* eradications on New Zealand's offshore islands) [46] compared to costs for the ongoing management of large, widespread alien bird populations at mainland locations (e.g., house crows *Corvus splendens* in East Africa) [47].

Damage costs on islands are also lower than damage costs at mainland locations, perhaps because there are fewer opportunities for alien birds to cause specific types of damage. Indeed, there were no reports of costs associated with damage to facilities/infrastructure/ buildings on islands, and a greater proportion of overall costs on islands were associated with management rather than damage when compared to mainland locations. However, although reported damage costs are relatively low on islands, they might be high relative to the income of local communities on those islands, particularly in low-income regions [48]. In these cases, agricultural damage could adversely affect human well-being by compromising food security [37].

The low costs incurred when eradicating populations of alien birds on islands are likely to be one-off costs (temporary), albeit with ongoing costs associated with any post-invasion monitoring. The high costs associated with damage at mainland locations are more likely to be ongoing, where some widespread species (e.g., common starlings) are difficult to remove completely. Indeed, the Western Australian Government has developed a long-term surveillance program to prevent the establishment of common starlings in the region [49], presumably recognising that the eradication of this species in Australia is unlikely. Because management costs to eradicate alien birds are typically lower than their damage costs (hypothesis 5), there could be economic benefits associated with early interventions to prevent the establishment of alien birds, particularly at mainland locations [50].

Alien birds are present on many islands around the world [10], where they can have damaging environmental impacts [45]. Costs associated with their management are likely to be incurred in the future, such as those required for the planned eradication of the Australian masked owl *Tyto novaehollandiae* from Lord Howe Island (Australia) [51]. Furthermore, although eradication costs tend to be one-off (temporary), they often require sustained effort,

imposing a financial burden over several years. For example, the eradication of common mynas on Denis Island (Seychelles) took place in three phases over five years [52]. The eradication of house crows on Zanzibar has so far failed, in part due to a lack of funding for the length of time needed to eradicate this species [47].

A few cost reports are historical, such as damage to agriculture by the Eurasian blackbird *Turdus merula* in Tasmania during the first half of the last century [53], but most are from 2000 to present. The sharp increase in reports since 2010 might arise because some alien bird species (particularly monk parakeets *Myiopsitta monachus* and ring-necked parakeets *Psittacula krameri*) are rapidly becoming more widespread [54], and are being managed with local-scale actions. For example, between 2015 and 2018 there were 75 different cost reports for the management of monk parakeets at 34 different sites in Catalonia, Spain. These local-scale actions tend to have low costs (although they are annual, ongoing costs in Spain) perhaps explaining why total yearly costs for birds as an entire taxonomic group have not increased much over the same period, and why average costs are falling. While these local-scale actions might be effective in addressing local-scale impacts, the successful eradication of alien parrots in Europe will require a coordinated, international response [55, 56]. Another reason why total costs are not rising is that some costly management actions to eradicate widespread species perceived to have the most damaging impacts have now been completed, such as the eradication of African sacred ibises *Threskiornis aethiopicus* in Europe [57].

Over half of all costs were for street pigeons. They damage buildings with corrosive droppings [58], cause flooding by blocking gutters with droppings and feathers [59], and cover structures with droppings. Maintenance to manage their impacts can be costly; the removal of 350 tonnes of droppings from a bridge in Canada cost US$640,000 [60]. Street pigeons also consume and spoil agricultural produce with droppings [61], and their presence at airports, where they are a risk to airline safety, requires costly exclusion and monitoring activities [62]. This human-commensal species thrives in many cities, and accurately calculating its global costs is challenging because many of its impacts are not reported, such as damage to residential dwellings. In addition, these reported costs are likely to be underestimated given that they have been calculated for only three countries (Canada, UK, and USA).

The higher costs caused by perching birds (Passeriformes) tend to result from agricultural damage. On Zanzibar Island (Tanzania), house crows consume crops, hinder aquaculture operations by stealing bait and pulling bungs out of boats, and attack and kill poultry and young livestock [63]. They also aggressively compete with, and prey on native bird species [64]. Unsuccessful attempts to eradicate the large population of house crows on Zanzibar (> 1 million individuals) [47] have so far cost > US$1.5 million [65]. However, a population of around 50,000 individuals was successfully eradicated in Singapore, costing US$765,000 [66], and a smaller population of 30 individuals was successfully eradicated from the island of Socotra (Yemen), costing US$20,500 [67].

Four waterfowl species (Anseriformes) have reported costs. However, most costs are associated with the control of ruddy ducks *Oxyura jamaicensis* in Western Europe to prevent hybridisation with native white-headed ducks *Oxyura leucocephala* [68]. The alien population of ruddy ducks was large and distributed across several countries; the ongoing control program has taken several years, and so far cost > US$28 million. Expensive control of large house crow and ruddy duck populations demonstrates that timely interventions to control alien bird populations at early stages of invasions (when they are relatively small) can prevent spiralling management costs [39, 69].

Two parrot species (Psittaciformes) with the largest alien ranges of all parrot species have reported costs: monk parakeets and rose-ringed parakeets. Introduced to many regions of the world [70, 71], their broad alien distribution provides these species with greater opportunity to

cause negative impacts when compared to alien parrot species with smaller ranges [33, 39]. Monk parakeets nest on electrical structures causing power outages [72]. They are an agricultural pest in their native range and are predicted to become a serious pest to agriculture in the Mediterranean if their population continues to grow [73]. They are also a nest-site facilitator, hosting other species of birds (both native and alien) [74, 75]. Rose-ringed parakeets damage agriculture, including almond orchards in Rome [76] (they are also an agricultural pest in their native range) [77], and they compete with native species for nest sites, which has caused declines in populations of Eurasian nuthatches in Belgium [6] and a threatened bat (greater noctule *Nyctalus lasiopterus*) in Spain [78]. In the UK, the rose-ringed parakeet population is increasing in abundance and range [71, 79], as are populations of this species and monk parakeets in Spain [54]. It is likely that costs associated with damage caused by these species and any future actions to manage their impacts are also increasing [69].

Costs have been reported for two landfowl species (Galliformes)—agricultural damage by common pheasants *Phasianus colchicus* in Germany [80] and management costs on Santa Cruz Island (USA) incurred to eradicate wild turkeys *Meleagris gallopavo* due to their negative impacts on biodiversity [81]. Such impacts have also been the catalyst for the eradication of African sacred ibises (Pelecaniformes) in mainland France [57] and weka (Gruiformes) on many of New Zealand's offshore islands [46].

Many costs assigned to the 'diverse/unspecified' category have been incurred in Australia and are associated with agricultural damage, particularly by frugivorous perching birds that consume soft fruit [82], including common starlings, Eurasian blackbirds, common mynas, and house sparrows [83]. Costs assigned to these species are likely to be higher than reported [84]. Other alien bird species with reported environmental and socio-economic impacts [14, 15] do not have reported costs (e.g., rooks *Corvus frugilegus* in New Zealand), which also suggests that costs associated with alien birds are underreported. Furthermore, costs incurred to control and eradicate alien birds are often not published, including costs for the eradication of house sparrows in Mauritius [85] and street pigeons in the Galapagos [86].

## Conclusions

Our review reveals that the economic costs of alien birds are widespread, but also underreported. Just 5% of established alien bird species have data describing their economic costs, and there are regions of the world occupied by many alien bird species where no costs were identified. However, it is likely that many species with no cost data have low, or no, economic costs. Indeed, the characteristics of alien birds and the nature of their impacts is most likely why their costs are lower than other taxonomic groups so far assessed using *InvaCost*. Nevertheless, some damaging impacts caused by alien birds, such as native species extirpations on islands, are difficult to monetise. Furthermore, some increasingly widespread alien bird species do have high costs (and damaging biodiversity impacts). Their eradication will require a coordinated, international response, but they are being managed with local-scale actions. Further research to understand why the impacts of alien birds tend to be less costly than those caused by other taxonomic groups of alien species could inform measures to reduce the costs associated with biological invasions. Indeed, our study demonstrates that we have much to learn about the economic costs of alien birds. Avenues for future research could include identifying characteristics of alien bird species that cause them to have high economic costs, and factors that cause variation in the distribution of costs across different regions. This research could help to predict the types of species that have the most damaging economic impacts, which could inform biosecurity measures to prevent their introduction as aliens.

## Supporting information

**S1 Fig. The spatial distribution of the observed economic costs associated with alien birds, with costs for street pigeons excluded (US\$, 2017 exchange rate).** k = thousand; m = million; bn = billion. $D_{ag}$ = costs associated with damage to agriculture; $D_{fa}$ = costs associated with damage to facilities/infrastructure/buildings; M = cost associated with management. This map was made with Natural Earth. Free vector and raster map data @ naturalearthdata. com.
(DOCX)

**S1 Table. Alien bird species with no data on economic costs that have been assessed as having 'harmful' environmental impacts under the _Environmental Impact Classification for Alien Taxa_ (EICAT) framework (Blackburn et al. 2014).**
(DOCX)

## Author Contributions

**Formal analysis:** Thomas Evans.

**Methodology:** Thomas Evans.

**Visualization:** Thomas Evans, Anna Turbelin.

**Writing – original draft:** Thomas Evans.

**Writing – review & editing:** Thomas Evans, Elena Angulo, Corey J. A. Bradshaw, Anna Turbelin, Franck Courchamp.

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
