## [Decision Letter · Decision Letter 0]

26 Jun 2023

PONE-D-23-06753Global economic costs of alien birdsPLOS ONE

Dear Dr. Evans,

Thank you for submitting your manuscript to PLOS ONE. After careful consideration, we feel that it has merit but does not fully meet PLOS ONE’s publication criteria as it currently stands. Therefore, we invite you to submit a revised version of the manuscript that addresses the points raised during the review process. The referees found some important points the authors are requested to improve, before letting this manuscript to proceed further in the revision process. For instance, the authors should better clarify where they obtained the data for the study. In addition, some statistical analyses should be provided to test the study hypotheses.

We look forward to receiving your revised manuscript.

Kind regards,

Mirko Di Febbraro

Academic Editor

PLOS ONE

Journal Requirements:

This work was supported by the French National Research Agency [ANR-18-EBI4-0004-07, 2018]; the BNP-Paribas Foundation Climate Initiative [2014-00000004292, 2014]; the AXA Research Fund Chair of Invasion Biology [2019]; and the AlienScenario project funded by BiodivERsA and Belmont-Forum call 2018 on biodiversity scenarios [I 4011-B32, 2018].

This work was supported by the French National Research Agency [ANR-18-EBI4-0004-07, 2018]; the BNP-Paribas Foundation Climate Initiative [2014-00000004292, 2014]; the AXA Research Fund Chair of Invasion Biology [2019]; and the AlienScenario project funded by BiodivERsA and Belmont-Forum call 2018 on biodiversity scenarios [I 4011-B32, 2018].

However, funding information should not appear in the Acknowledgments section or other areas of your manuscript. We will only publish funding information present in the Funding Statement section of the online submission form. 

This work was supported by the French National Research Agency [ANR-18-EBI4-0004-07, 2018]; the BNP-Paribas Foundation Climate Initiative [2014-00000004292, 2014]; the AXA Research Fund Chair of Invasion Biology [2019]; and the AlienScenario project funded by BiodivERsA and Belmont-Forum call 2018 on biodiversity scenarios [I 4011-B32, 2018].

4. We note that Figure 1 and S1 in your submission contain map images which may be copyrighted. All PLOS content is published under the Creative Commons Attribution License (CC BY 4.0), which means that the manuscript, images, and Supporting Information files will be freely available online, and any third party is permitted to access, download, copy, distribute, and use these materials in any way, even commercially, with proper attribution. For these reasons, we cannot publish previously copyrighted maps or satellite images created using proprietary data, such as Google software (Google Maps, Street View, and Earth). For more information, see our copyright guidelines: http://journals.plos.org/plosone/s/licenses-and-copyright.

a. You may seek permission from the original copyright holder of Figure 1 and S1 to publish the content specifically under the CC BY 4.0 license.  

Reviewers' comments:

Reviewer's Responses to Questions

**Comments to the Author**

1. Is the manuscript technically sound, and do the data support the conclusions?

Reviewer #1: Partly

Reviewer #2: Partly

2. Has the statistical analysis been performed appropriately and rigorously? 

Reviewer #1: N/A

Reviewer #2: No

3. Have the authors made all data underlying the findings in their manuscript fully available?

Reviewer #1: No

Reviewer #2: Yes

4. Is the manuscript presented in an intelligible fashion and written in standard English?

Reviewer #1: Yes

Reviewer #2: Yes

5. Review Comments to the Author

Reviewer #1: In general, the introduction shows too much very specific (often local) examples and too few general assumptions in my opinion. Please increase literature search, avoid describing specific examples and increase the number of cited works (for instance, take a look at alien parakeets in Europe).

Still I am confused: does InvaCOst include also data on alien birds or, as you said at lines 75-76, alien birds are still excluded? If so, I do not understand the novelty of this work. If not, I do not understand how did you get your data. This is an important issue to be clarified, before I can provide reliable comments on the rest of the MS.

Furthermore, I would like to suggest authors to discuss more about Figure 1, as in Europe several countries are not controlling alien birds, but they are spending for their damages, and these costs are not quantified.

Lines 17-18. Do you refer to worldwide costs? Please clarify.

Keywords. You never mentioned Columba livia as an invasive species in the abstract. Please clarify or try to find different keywords.

Lines 42-43. This is a very specific example. It could be ok, but you have to look at different examples also, as being in the introductive part.

Line 46. Please, change “demonstrates” with “shows”. You have no experiment in your manuscript.

Figure 2. Please, scientific names should be in italics.

Reviewer #2: The paper examines the distribution of economic costs from alien birds over space, time and species. This may help understand the reason for the cost and may help to reduce economics costs of biological invasions.

The data size is however limited with only 351 cost records (333 costs with high reliability) spanning from 1940 -2020. Data for costs from are unavailable for some species and regions. There are also not statistical tests of the hypotheses (some due to lack of data). The data analysis of the cost distribution and underlying mechanism is mainly descriptive. Here are some suggestions to improve the paper:

1 In the regions with reported data, it will be helpful to run a set of regression analysis to see how regions, species, taxonomic groups, impact type (environmental impact data from Reference 20 ), or year fixed effect can explain the variation of the economic costs. This provides stronger tests of the 6 hypotheses.

2 Find some proxy of the economic cost especially in the unreported regions. The paper states that economics costs are likely to be underreported because.:

1) Due to budget constraints, there is no monitoring/research in some regions, particularly low-income regions.

2) Within the reported regions, only costs associated with the most damaging species are reported. The other species may lead to relatively low costs.

It may be possible to use the data from the reported region to infer information about economic costs in the unreported regions. For example, if we know the reason for the invasive species and if one species appears in both reported region(s) and unreported region, can we assume the economics costs in the unreported region is proportional to the economics costs in the reported regions? The proportion can be a function of regional GDP, year, etc.

3The paper also states that some impacts are hard to monetize. So, no cost data even when the environmental impact is reported high.

It can also be helpful to construct a model to predict economic cost based on environmental and social impacts and regional characteristics. It would be interesting to explain the gap between the predicted economic cost and reported cost.

6. PLOS authors have the option to publish the peer review history of their article (what does this mean?). If published, this will include your full peer review and any attached files.

Reviewer #1: **Yes: **Emiliano Mori

Reviewer #2: No

---

## [Author Response · Author response to Decision Letter 0]

12 Jul 2023

We thank both reviewers for their helpful comments, to which we have responded in the attached 'response to comments' document. Our responses are highlighted in blue text. We have highlighted changes to the revised manuscript in yellow, and provided line number references in our responses.

We hope that our manuscript is now suitable for publication.

Best regards

Thomas Evans

---

## [Decision Letter · Decision Letter 1]

31 Jul 2023

PONE-D-23-06753R1Global economic costs of alien birdsPLOS ONE

Dear Dr. Evans,

Thank you for submitting your manuscript to PLOS ONE. After careful consideration, we feel that it has merit but does not fully meet PLOS ONE’s publication criteria as it currently stands. Therefore, we invite you to submit a revised version of the manuscript that addresses the points raised during the review process. One of the two reviewers still requires important amendments to reach the manuscript full potential. Specifically, a more robust support to some of the study hypotheses is requested.

We look forward to receiving your revised manuscript.

Kind regards,

Mirko Di Febbraro

Academic Editor

PLOS ONE

Reviewers' comments:

Reviewer's Responses to Questions

**Comments to the Author**

1. If the authors have adequately addressed your comments raised in a previous round of review and you feel that this manuscript is now acceptable for publication, you may indicate that here to bypass the “Comments to the Author” section, enter your conflict of interest statement in the “Confidential to Editor” section, and submit your "Accept" recommendation.

Reviewer #1: All comments have been addressed

Reviewer #2: (No Response)

2. Is the manuscript technically sound, and do the data support the conclusions?

Reviewer #1: Yes

Reviewer #2: Partly

3. Has the statistical analysis been performed appropriately and rigorously? 

Reviewer #1: Yes

Reviewer #2: No

4. Have the authors made all data underlying the findings in their manuscript fully available?

Reviewer #1: Yes

Reviewer #2: Yes

5. Is the manuscript presented in an intelligible fashion and written in standard English?

Reviewer #1: Yes

Reviewer #2: Yes

6. Review Comments to the Author

Reviewer #1: Authors addressed all of my previous comments. Therefore, the MS can be accepted for publication.

All the best.

Reviewer #2: The paper investigates the distribution of economic costs arising from alien birds, focusing on spatial, temporal, and species variations. The primary approach involves a descriptive analysis of the cost distribution and underlying mechanisms. While the research provides valuable insights, there are some aspects that require clarification and additional support to strengthen the overall findings.

1 Support for Hypothesis 3 (more cost data are available for damaging alien bird species because they are more likely to be managed.)

While the paper mentions that 60% of alien bird species with costs have harmful environmental impacts, it would be more informative to provide a summary of the percentage of each environmental impact category under the Environmental Impact Classification for Alien Taxa (EICAT) framework) to understand the severity of their effects accurately.

2 Support for Hypothesis 4 (data on costs will be unavailable for some regions where severe environmental impacts are reported as environmental impacts are hard to monetize.)

The evidence supporting Hypothesis 4, which suggests that data on costs may be unavailable in regions with severe environmental impacts, lacks substantial support from the data. Calculating the percentage of alien bird species with damaging costs among those with "harmful" or higher impacts would offer more compelling evidence for this hypothesis.

3 Conflicting Evidence for Hypothesis 5

The paper introduces Hypothesis 5, stating that management costs for eradicating alien birds are typically low compared to their damage costs. However, there is conflicting evidence presented in the figures (Figure S4 and Figure S2). In Figure S4, damage cost dominates management costs (with and without street pigeons). But in Figure S2, you have higher percentage of management costs than damage costs when street pigeons are included. Moreover, it would be helpful if you refer to the figure that support your hypothesis.

4 Statement in the abstract and Support for Hypothesis 6.

In the Abstract (Line 24), the paper states that "Research to understand why (alien birds tend to have lower costs than many other alien taxa (Hypothesis 6)) may inform measures to reduce costs associated with biological invasions".

With limited and potentially underreported data, it is hard to justify hypothesis 6. Moreover, the paper discusses the two following reasons of why alien birds have lower costs than the other taxa, but there is no concrete evidence to support these mechanisms:

a). Lower costs for alien birds may come from underestimation:

The paper acknowledges that costs for alien birds may be underestimated due to underreported environmental costs that are challenging to monetize. Additionally, minor environmental and socio-economic impacts of alien birds have led to limited research. These points provide valuable insights into potential reasons for the lower perceived costs of alien birds. However, it would be beneficial to include specific examples or case studies to illustrate the extent of underestimation and its impact on cost assessments.

b). Mechanisms for Lower Costs in Alien Birds than other taxa:

The paper suggests possible mechanisms that could explain why alien birds tend to have lower costs compared to other taxa, such as their habitat preference (above ground or water level). Nevertheless, the paper lacks concrete evidence to support these mechanisms. For example, there is no information provided to demonstrate whether the environmental impacts of alien birds are indeed smaller than those of ants and crayfish. Low environmental costs of alien birds might lead to low costs. However, together with costs underestimation, high environmental impacts of alien birds may end up with lower costs reflected by data. Including empirical data or relevant studies to validate these hypotheses would strengthen the paper's argument.

5 Is Inva cost the only dataset that provides global information about cost of alien birds? Is there any other dataset available even locally? Other relevant data will be able to support the arguments.

Overall, the paper provides valuable insights into the distribution of economic costs associated with alien birds, considering spatial, temporal, and species-related variations. However, the descriptive nature of the data analysis limits the paper's ability to draw robust conclusions. By incorporating specific examples, empirical data, and further support for the hypothesis and proposed mechanisms, the authors can enhance the paper's impact and provide more practical implications for managing biological invasions linked to alien bird species.

7. PLOS authors have the option to publish the peer review history of their article (what does this mean?). If published, this will include your full peer review and any attached files.

Reviewer #1: **Yes: **Emiliano Mori

Reviewer #2: No

---

## [Author Response · Author response to Decision Letter 1]

24 Aug 2023

Thank you very much for the additional comments on our manuscript. We have prepared a 'response to comments' document, providing a detailed response to each of these comments. We hope that our manuscript is now acceptable for publication.

---

## [Decision Letter · Decision Letter 2]

15 Sep 2023

PONE-D-23-06753R2Global economic costs of alien birdsPLOS ONE

Dear Dr. Evans,

Thank you for submitting your manuscript to PLOS ONE. After careful consideration, we feel that it has merit but does not fully meet PLOS ONE’s publication criteria as it currently stands. Therefore, we invite you to submit a revised version of the manuscript that addresses the points raised during the review process.

 Some minor amendments are still required by the referee before accepting the manuscript for publication.

We look forward to receiving your revised manuscript.

Kind regards,

Mirko Di Febbraro

Academic Editor

PLOS ONE

Journal Requirements:

Reviewers' comments:

Reviewer's Responses to Questions

**Comments to the Author**

1. If the authors have adequately addressed your comments raised in a previous round of review and you feel that this manuscript is now acceptable for publication, you may indicate that here to bypass the “Comments to the Author” section, enter your conflict of interest statement in the “Confidential to Editor” section, and submit your "Accept" recommendation.

Reviewer #2: All comments have been addressed

2. Is the manuscript technically sound, and do the data support the conclusions?

Reviewer #2: (No Response)

3. Has the statistical analysis been performed appropriately and rigorously? 

Reviewer #2: (No Response)

4. Have the authors made all data underlying the findings in their manuscript fully available?

Reviewer #2: Yes

5. Is the manuscript presented in an intelligible fashion and written in standard English?

Reviewer #2: Yes

6. Review Comments to the Author

Reviewer #2: Thank you for addressing my comments. I only have 1 question now.

1 Reliability of Street Pigeon Cost Data:

1) Dataset Information (lines 136-137):

The dataset offers descriptions of the reliability for cost records, categorizing them as 'low' or 'high'.

2) Reference from lines 172-173:

A significant portion (95%) of the cost records (n=333) are marked as 'high' reliability. It's unclear if this percentage includes the costs associated with street pigeons.

3) Reference from lines 163-164:

The costs related to street pigeons have been designated a 'low' reliability and are categorized under 'mixed' costs.

Concerns:

There is ambiguity about the reliability classification for street pigeon costs in the dataset, especially since they constitute over half of all costs (US $2 billion out of US $3.6 billion from reference line 169-171).

If the dataset classifies street pigeon costs as 'high reliability' but they are later marked as 'low', this discrepancy could affect the reliability of the overall cost data (US$3.6 billion). This is particularly important considering:

a) Street pigeons contribute to 50% of the total costs (US $2 billion).

b) The validity of the 'high' reliability label for other birds might be brought into question.

Recommendations:

Kindly provide clarity on the reliability rating assigned to street pigeon costs in the dataset.

It would also be beneficial to include an explanatory note for the "mixed" category (representing both management and damage) in relevant figures or tables.

7. PLOS authors have the option to publish the peer review history of their article (what does this mean?). If published, this will include your full peer review and any attached files.

Reviewer #2: No

---

## [Author Response · Author response to Decision Letter 2]

21 Sep 2023

Thank you for raising this additional point, which we have addressed in the attached 'response to comments' document. We appreciate your input, and we hope our manuscript is now suitable for publication.

Best regards

Thomas Evans

---

## [Editor Report · Decision Letter 3]

2 Oct 2023

Global economic costs of alien birds

PONE-D-23-06753R3

Dear Dr. Evans,

We’re pleased to inform you that your manuscript has been judged scientifically suitable for publication and will be formally accepted for publication once it meets all outstanding technical requirements.

Kind regards,

Mirko Di Febbraro

Academic Editor

PLOS ONE
---

## [Editor Report · Acceptance letter]

9 Oct 2023

PONE-D-23-06753R3 

Global economic costs of alien birds 

Dear Dr. Evans:

I'm pleased to inform you that your manuscript has been deemed suitable for publication in PLOS ONE. Congratulations! Your manuscript is now with our production department. 

Kind regards, 

on behalf of

Dr. Mirko Di Febbraro 

Academic Editor

PLOS ONE